# Eye-gaze Guided Multi-modal Alignment for Medical Representation Learning

**Chong Ma**[1], **Hanqi Jiang**[2], **Wenting Chen**[3], **Yiwei Li**[2], **Zihao Wu**[2]
**Xiaowei Yu**[4], **Zhengliang Liu**[2], **Lei Guo**[1], **Dajiang Zhu**[4], **Tuo Zhang**[1]
**Dinggang Shen**[5], **Tianming Liu**[2], **Xiang Li**[6*]
[1]Northwest Polytechnical University    [2]University of Georgia    [3]City University of Hong Kong
[4]University of Texas at Arlington    [5]ShanghaiTech University & Shanghai United Imaging Intelligence Co.
[6]Massachusetts General Hospital, Harvard University

## Abstract

In the medical multi-modal frameworks, the alignment of cross-modality features presents a significant challenge. However, existing works have learned features that are implicitly aligned from the data, without considering the explicit relationships in the medical context. This data-reliance may lead to low generalization of the learned alignment relationships. In this work, we propose the **E**ye-gaze **G**uided **M**ulti-modal **A**lignment (EGMA) framework to harness eye-gaze data for better alignment of medical visual and textual features. We explore the natural auxiliary role of radiologists' eye-gaze data in aligning medical images and text, and introduce a novel approach by using eye-gaze data, collected synchronously by radiologists during diagnostic evaluations. We conduct downstream tasks of image classification and image-text retrieval on four medical datasets, where EGMA achieved state-of-the-art performance and stronger generalization across different datasets. Additionally, we explore the impact of varying amounts of eye-gaze data on model performance, highlighting the feasibility and utility of integrating this auxiliary data into multi-modal alignment framework.

## 1 Introduction

With the development of multi-modal learning, pre-trained models can now utilize large amounts of paired multi-modal data, such as image-text pairs, audio-text pairs, etc., to optimize the multi-modal feature extraction and alignment capabilities. With the emergence of the CLIP [42] model, contrastive learning has become the prominent framework of multi-modal learning. The advantage of this framework lies in its simplicity of structure and it does not require sample-level annotations. However, the main drawback is its heavy reliance on the scale of training data. Subsequent works have optimized this framework by leveraging potential auxiliary information between image and text data. For instance, GLIP [32] and RegionCLIP [61] utilized pre-predicted annotation information to perform fine-grained region-level pre-training. They introduced detection networks firstly to predict image regions relevant to the text prompt, and then trained the model to align these image regions with their corresponding text descriptions. However, these models heavily rely on the performance of the ROI detector and have high computational complexity. FILIP [57] proposed a refined multi-modal alignment operation after the encoder, relying solely on image patches and text tokens. Although this further explores the local feature relationships between multi-modal data, it still requires sufficient data support. When training on small-scale datasets, especially in the medical field, accurately learning alignment features between modalities becomes more challenging [6, 60].

---

*Corresponding author: xli60@mgh.harvard.edu

38th Conference on Neural Information Processing Systems (NeurIPS 2024).

To address the scarcity of medical data, studies [4, 58] have introduced self-supervised training into the CLIP framework to further enhance encoder performance. Additionally, weak labels between images and texts have been incorporated during pre-training to aid multi-modal alignment [55]. Some studies [15, 53] utilized fine-grained alignment between chest image patches and text tokens for pre-training [57]. However, unlike natural images and text, the relationship between medical images and diagnostic text is often more complex and challenging to learn. Moreover, with insufficient data, models are prone to learning shortcut features unrelated to disease diagnosis, resulting in poor generalization ability [10, 36, 35]. Therefore, it is crucial to learn useful alignment information from relatively limited medical multi-model datasets.

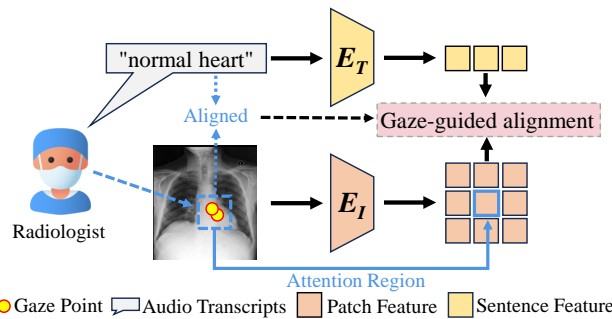

Figure 1: The guiding role of radiologists' eye-gaze data. The text provided by radiologists during diagnosis aligns naturally with the attention regions.

In this study, we fully explore the auxiliary role of eye-gaze data from radiologists in multi-model alignment. Eye-gaze data can intuitively reflect the image regions radiologists focus on, providing insights into their cognitive behavior during diagnosis [7]. Therefore, compared to refined annotations like bounding boxes and masks, eye-gaze data can also provide useful auxiliary information for the model [24, 54, 35]. Moreover, collecting eye-gaze data from radiologists during the diagnostic process is more time-efficient than annotating bounding boxes and masks [25, 35]. For the multi-modal medical dataset, EYE GAZE [22] and REFLACX [31] collected eye-gaze data from radiologists while diagnosing chest X-rays. Additionally, these datasets recorded synchronized voice data, where radiologists verbalized their diagnoses while observing the images. As shown in Fig. 1, we found that the radiologists' attention regions on the image naturally align with the diagnostic text over time. Therefore, we believe this type of eye-gaze data can provide expert prior knowledge for training the alignment between medical visual and textual features. Thus, considering the utilization of eye-gaze data to assist in multi-modal model training, we propose the **E**ye-gaze **G**uided **M**ulti-modal **A**lignment framework (EGMA). Our model first segments the transcribed text into individual sentences and obtains radiologists' attention heatmaps. Subsequently, we obtain encoded features of image patches and sentences through image and text encoders, generating instance-level similarity matrix. Then, we compute the loss between this matrix and the attention heatmaps, integrating refined feature representations for subsequent contrastive loss. To further leverage the assisting role of eye-gaze data in aligning images and texts, we combine the eye-gaze heatmaps with the similarity matrix derived from model, serving as weights to calculate cross-modality mapping loss. Experimental results on zero-shot classification and retrieval tasks reveal that our framework surpasses other leading methods in performance across diverse datasets and under multiple dataset size scenarios. Specifically, the EGMA framework yielded a remarkable 3.9% improvement in image-to-text matching tasks and an impressive 19.75% increase in text-to-image matching tasks. These results underscore the cutting-edge and efficacious nature of our approach, highlighting its substantial advancements over existing methodologies. We also explore the auxiliary effect of using eye-gaze data of different scales on the model, finding that even a small portion of eye-gaze data can enhance the model's multi-modal processing capability. Moreover, the fine-tuned classification results of EGMA achieved the best performance across multiple datasets. The code of this work is available on Github[2].

In summary, the main contributions of this work are as follows:

---

[2]https://github.com/MoMarky/EGMA

- We propose EGMA, a novel framework for medical multi-modal alignment, marking the first attempt to integrate eye-gaze data into vision-language pre-training.

- EGMA outperforms existing state-of-the-art medical multi-modal pre-training methods, and realizes notable enhancements in image classification and image-text retrieval tasks.

- EGMA demonstrates that even a small amount of eye-gaze data can effectively assist in multi-modal pre-training and improve the feature representation ability of the model.

## 2 Related Works

**Medical Vision-language Pre-training (Med-VLP):** In pursuit of Artificial General Intelligence (AGI), Vision-language Pre-training (VLP) has become a pivotal area in AI research. The advent of the transformer architecture [51] has significantly accelerated progress in the multi-modal domain by integrating vision and language, with VLP frameworks focusing on fusion encoders that use cross-attention mechanisms to amalgamate visual and textual features [34, 47]. The introduction of CLIP [42] marked a breakthrough, leading to numerous CLIP-based VLP frameworks incorporating contrastive loss as a core component [57, 32]. In the medical field, ConVIRT [59] serves as an equivalent to CLIP, while MedCLIP [55] addresses the challenge of insufficient paired image-text data by integrating knowledge extraction techniques. BioViL [2] enhances performance through specialized biomedical text BERT encoders in contrastive learning tasks. GLoRIA [15] proposes multi-modal global-local representation learning, and MGCA [53] introduces alignment at pathological region, instance, and disease levels. Furthermore, study [56] incorporates knowledge bases to infuse expert medical knowledge into the system.

**Eye-tracking Technology in Radiology:** In medical imaging diagnostics, eye-tracking technology has proven valuable for decades, revealing how experienced radiologists can quickly identify hidden lesions through comprehensive observation [28, 29, 7, 26]. Integrating radiologists' eye-gaze data with deep learning models has significantly advanced the field. For instance, merging eye-gaze data with Convolutional Neural Networks (CNNs) has enhanced lesion detection accuracy [24], and visual search patterns in mammography have linked human visual attention with CNN performance [37]. Comprehensive datasets combining eye-gaze data and disease diagnoses have facilitated multi-task processing [21], while attention consistency modules have improved CNN accuracy in diagnosing osteoarthritis from knee X-rays [54]. Recently, integrating eye-gaze data with Vision Transformer (ViT) models has further pushed the boundaries of medical image processing [35]. Additionally, multi-modal guidance systems replicating eye tracking and probe manipulation in ultrasound examinations have significantly enhanced scanning accuracy and efficiency [38]. Despite these advancements, fully integrating eye-gaze data with image-text alignment strategies in medical vision-language models remains an ongoing research challenge.

## 3 Method

As shown in Fig. 2, the framework of our proposed method consists of four main components. Firstly, we extract features from image and text in part $A$ to obtain a refined instance-level similarity matrix. Secondly, in part $B$, we integrate textual transcripts derived from radiologists' audio, images, and eye-gaze data, to visualize and map radiologists' attention onto specific regions of images during diagnosis. This process establishes alignment between texts and images, facilitating model training. The detailed gaze data processing methods are described in Sec. 3.1. Given that eye-gaze data tightly links textual and localized visual information, after obtaining auxiliary information from part $B$, we introduce eye-gaze guided refined alignment training strategies, as depicted in Parts $C$ and $D$ of Fig. 2. Specifically, we introduce the optimization algorithm for eye-gaze guided fine-grained text-image similarity matrix in Part $C$ in Sec. 3.2. Finally, in Sec. 3.3, we present the algorithm for eye-gaze guided cross-modality mapping.

### 3.1 Multi-modal Data Processing

With the development of data collection technologies such as eye-tracking and speech recognition, it has become possible to collect and process multi-modal interaction data of radiologists during the diagnostic process. In this work, we utilize MIMIC-EYE [14] datasets as our training set, consisting of 3689 images extracted from the MIMIC datasets [19, 20, 18, 17]. Each sample is accompanied

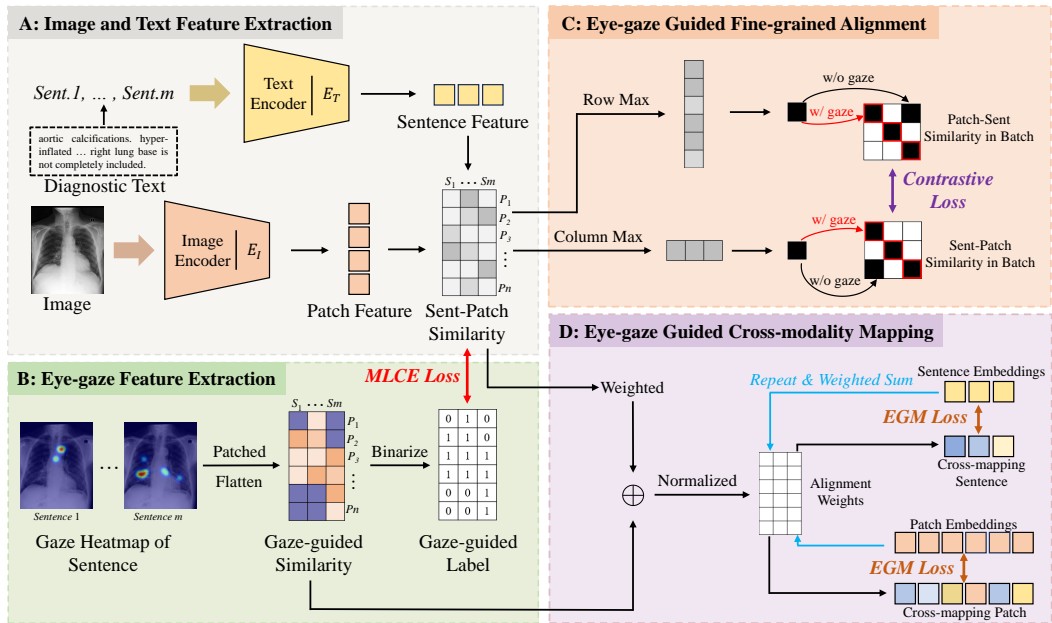

Figure 2: The framework of EGMA. After images and text are processed by the encoder in Part $A$, patch feature and sentence feature representations are obtained, resulting in a fine-grained similarity matrix for instances. Subsequently, the two types of eye-gaze-based auxiliary information obtained in Part $B$ are used for fine-grained and cross-mapping alignment in Part $C$ and Part $D$, respectively.

by corresponding eye-tracking data and transcripts text. These eye-tracking data are provided by the publicly available EYE GAZE [22] and REFLACX [31] datasets on PhysioNet [12]. Since each modality is synchronized, the audio data is aligned with the eye-gaze data in time. By segmenting the audio based on the time before and after the pronunciation of each word, we can align the transcripts with the audio, thereby aligning sentence-level text with eye-gaze data. Subsequently, we generate attention heatmap based on eye-gaze data and images to represent the image regions the radiologist focuses on. Through the aforementioned data processing steps, we achieve precise alignment between sentence-level text and image regions. Detailed processing method of eye-gaze and audio transcripts can be found at Appendix D.

### 3.2 Eye-gaze Guided Fine-grained Alignment

The core idea of contrastive learning is to bring the features of related samples closer while pushing away the features of unrelated samples. During the training progress of CLIP [42] model, assuming a batch size of $b$ and input data $\{x_k^I, x_k^T\}$ $(k = 1, \cdots, b)$ representing image-text pairs, global features $z_k^I = E_I(x_k^I) \in \mathbb{R}^{1 \times d}$ and $z_k^T = E_T(x_k^T) \in \mathbb{R}^{1 \times d}$ are obtained through image encoder $E_I$ and text encoder $E_T$. Subsequently, the cosine similarity $s_{k,l}^{I2T}$ and $s_{k,l}^{T2I}$ between the two modalities is computed, with the following formula:

$$s_{k,l}^{I2T} = cossim(z_k^I, z_l^T), \ s_{k,l}^{T2I} = cossim(z_k^T, z_l^I) \quad 1 \leq l \leq b \tag{1}$$

where $cossim$ is cosine similarity, $s_{k,l}^{I2T}$ is the image-to-text similarity, $s_{k,l}^{T2I}$ is the text-to-image similarity, and $l$ is the index number of the another modality. Then, the image-to-text contrastive loss $L_k^{I2T}$ for $x_k^I$ and text-to-image contrastive loss $L_k^{T2I}$ for $x_k^T$ can be formulated as:

$$L_k^{I2T}(x_k^I, \left\{x_l^T\right\}_{l=1}^b) = -\frac{1}{b}log\frac{exp(s_{k,k}^{I2T}/\tau)}{\sum_l(exp(s_{k,l}^{I2T}/\tau))}, \ L_k^{T2I}(x_k^T, \left\{x_l^I\right\}_{l=1}^b) = -\frac{1}{b}log\frac{exp(s_{k,k}^{T2I}/\tau)}{\sum_l(exp(s_{k,l}^{T2I}/\tau))} \tag{2}$$

where $\tau$ is a learned temperature. It is worth noting that in the calculation of the loss mentioned above, both the image and text utilize global-level features, while the auxiliary information generated from eye-gaze data emphasizes the local-level features between modalities. Therefore, based on [57], we replace instance feature $z_k^I$ and $z_k^T$ with $P_k^n \in \mathbb{R}^{n \times d}$ and $S_k^m \in \mathbb{R}^{m \times d}$, where $P_k^i(1 \leq i \leq n)$

is the $i$-th patch feature of $x_k^I$ and $S_k^j (1 \le j \le m)$ is the $j$-th sentence feature of $x_k^T$, and $n, m$ are the image patch number and the sentence number of report. Then we calculate the similarities of sentence-to-patch $x_k^{S2P} \in \mathbb{R}^{m \times n}$ and patch-to-sentence $x_k^{P2S} \in \mathbb{R}^{n \times m}$ in one instance:

$$x_k^{S2P} = cossim(S_k^j, P_k^i), \; x_k^{P2S} = cossim(P_k^i, S_k^j) \tag{3}$$

For each heatmap corresponding to a sentence, we initially divide it into $n$ patches similar to the image. Subsequently, we concatenate the heatmaps of $m$ sentences to obtain the Gaze-guided Similarity matrix $GS_k$ for input $\{x_k^I, x_k^T\}$ (as illustrated in Fig. 2.B). In this matrix, non-zero elements indicate the semantic correlation between the corresponding sentences and image patches. Thus, we binarize $GS_k$, setting non-zero regions to 1, resulting in the Gaze-guided Label matrix $GL_k$. After this step, we compute the multi-label cross-entropy (MLCE) loss for $x_k^{S2P}$ and $x_k^{P2S}$, completing the optimization for fine-grained alignment between positive sample pairs $\{x_k^I, x_k^T\}$, as follows:

$$fL_k^{S2P} = mlce(x_k^{S2P}, GL_k), \; fL_k^{P2S} = mlce(x_k^{P2S}, (GL_k)^{\mathrm{T}}) \tag{4}$$

where $mlce$ is the multi-label cross-entropy loss. Subsequently, we calculate the fine-grained features $\hat{z}_k^I$ and $\hat{z}_k^T$ as follows:

$$\hat{z}_k^I = \frac{1}{n} \sum_{i=1}^{n} \max_j [(x_k^{P2S})_{ij}], \; \hat{z}_k^T = \frac{1}{m} \sum_{j=1}^{m} \max_i [(x_k^{S2P})_{ji}] \tag{5}$$

Then, we replace the $z_k^I, z_k^T$ with the updated $\hat{z}_k^I, \hat{z}_k^T$ in Eq. 1. Finally, the fine-grained image-to-text loss $\hat{L}_k^{I2T}$ and text-to-image loss $\hat{L}_k^{T2I}$ are computed based on Eq. 2. The formula for our Eye-gaze Guided Fine-grained (EGF) alignment loss is as follows:

$$L_{EGF} = \frac{1}{2b} \sum_{k=1}^{b} (fL_k^{S2P} + fL_k^{P2S}) + \frac{1}{2} \sum_{k=1}^{b} (\hat{L}_k^{T2I} + \hat{L}_k^{I2T}) \tag{6}$$

### 3.3 Eye-gaze Guided Cross-modality Mapping

In the previous section, we replaced the global instance logits in the traditional batch clip loss with fine-grained instance logits that consider local features and optimized the alignment between these local features using gaze information. The text in our work is recorded by radiologists while observing images, implying a close semantic relationship between the focus region and the corresponding text. To further optimize the alignment between modalities, we continue to incorporate eye-gaze data assistance into the cross-modality mapping process. In this work, we first utilize matrices $GS_k$, $x_k^{P2S}$ and $x_k^{S2P}$ to generate the image-to-text and text-to-image alignment weight matrix $W^{I2T} \in \mathbb{R}^{n \times m}$ and $W^{T2I} \in \mathbb{R}^{m \times n}$. The calculation formula is as follows:

$$W^{I2T} = norm(\omega(x_k^{P2S}) + GS_k), \; W^{T2I} = norm(\omega(x_k^{S2P}) + (GS_k)^{\mathrm{T}}) \tag{7}$$

where $norm$ is normalization and $\omega$ consists of sparse and binarize operations. After obtaining the weight matrix, we perform the mapping from text features $S_k^m$ to image features $Cross\_P_k^n \in \mathbb{R}^{n \times d}$ and from image features $P_k^n$ to text features $Cross\_S_k^m \in \mathbb{R}^{m \times d}$ according to the following formula:

$$Cross\_P_k^i = \sum_{j=1}^{m} S_k^j \cdot W_{ij}^{I2T}, \; Cross\_S_k^j = \sum_{i=1}^{n} P_k^i \cdot W_{ji}^{T2I} \tag{8}$$

where $i \in [1, n]$ is the $i$-th patch feature of $P_k^n$ and $j \in [1, m]$ is the $j$-th sentence feature of $S_k^m$. Subsequently, we use the mapped features along with the target features as inputs to compute the alignment contrastive loss defined in Eq. 2, obtaining the image mapping loss $mL_k^I$ and the text mapping loss $mL_k^T$. The formula for our Eye-gaze Guided cross-model Mapping (EGM) loss is as follows:

$$L_{EGM} = \frac{1}{2} \sum_{k=1}^{b} (mL_k^I + mL_k^T) \tag{9}$$

Finally, the total loss of our model within a batch is $L = L_{EGF} + L_{EGM}$. In our training process, considering the proportion of eye-gaze data, batches may contain both types of data. When encountering samples without eye-gaze data, the EGF module does not compute the loss from Eq. 4, and the weight matrix in the Eq. 7 of EGM module also excludes the $GS_k$.

# 4 Experiments

In this study, we first conduct supervised and zero-shot classification as well as zero-shot retrieval experiments in Sec. 4.1 to validate the model's generalization performance and its representation capability of multi-modal features. Then, in Sec. 4.2, we perform ablation studies on various modules of EGMA. Additionally, to further investigate the auxiliary effect of eye-gaze data, we compare the performance when guided by different amounts of eye-gaze data. Finally, in Sec. 4.3, we visualize the model's feature representations and the learned image-text relationships, further demonstrating the model's performance and interpretability.

## 4.1 Comparison with State-of-the-Arts

**Image Classification** We conduct supervised classification experiments on the CheXpert [16], RSNA [44], and SIIM-ACR [45] datasets. CheXpert [16] is a large-scale public dataset for chest radiograph interpretation, it comprises 224,316 chest radiographic images. Following [53], we utilize the official training split as our training set, and the official validation set of 202 images with expert-label as our test set. RSNA [44] is a comprehensive dataset for Pneumonia diagnosing. It contains 29,700 chest X-ray images categorized into $normal$ and $pneumonia$ positive category. We follow [53] to divide the data into 70% for training, 15% for validation, and 15% for testing. SIIM-ACR [45] is a chest dataset used for pneumothorax diagnosing. It consists of 2379 images with pneumothorax and 8300 images without pneumothorax. In this work, we utilize a subset defined in [43] as our test set, with the remaining data used for training and validation. More details of dataset can be found in the Appendix C.

In the supervised classification experiments, we adopt the linear classification settings [15], where the pre-trained image encoder is frozen, and only a randomly initialized linear classification head is trained. We adopt area under ROC curve (AUROC) metric to evaluate all model's performance. And for better validate the model's efficiency, we test its performance using 1%, 10%, and 100% of the training set. As shown in Tab. 1, our model achieved the best results compared to other models. Additionally, with only 1% of the training set, our model outperformed the second-best model by 1.11%, 0.8%, and 1.94% on the CheXpert, RSNA, and SIIM-ACR datasets, respectively. Moreover, as the amount of training data increased, the model's performance improved significantly. This demonstrates that, with the assistance of radiologists' eye-gaze data, our model possesses strong multi-modal feature representation capabilities.

Table 1: Comparison results of supervised classification task with other SOTA models on CheXpert, RSNA, and SIIM-ACR datasets. Area under ROC curve (AUROC) is reported with different portions of training data: 1%, 10%, 100%. **Red** and blue denote the best and second-best results.

| Method | CheXpert[16] | | | RSNA [44] | | | SIIM-ACR [45] | | |
|---|---|---|---|---|---|---|---|---|---|
| | 1% | 10% | 100% | 1% | 10% | 100% | 1% | 10% | 100% |
| ConVIRT [59] | 85.90 | 86.80 | 87.30 | 77.40 | 80.10 | 88.60 | - | - | - |
| BioViL [2] | 81.95 | 85.37 | 88.62 | 81.76 | 85.68 | 88.64 | 80.26 | 82.79 | 90.51 |
| MedKLIP [56] | - | - | - | 87.31 | 87.99 | 89.31 | 85.27 | 90.71 | 91.88 |
| MGCA [53] | 85.80 | 87.66 | 89.30 | 85.22 | 87.54 | 89.24 | 86.12 | 89.66 | 92.16 |
| GLoRIA [15] | 86.60 | 87.80 | 88.10 | 86.10 | 88.00 | 88.60 | - | - | - |
| PRIOR [5] | 86.16 | 87.08 | 89.08 | 86.72 | 88.07 | 89.19 | 88.35 | 89.72 | 92.49 |
| MedCLIP [55] | 85.74 | 87.49 | 88.02 | 87.61 | 88.19 | 89.10 | 88.84 | 91.13 | 92.18 |
| EGMA(Ours) | **87.71** | **88.92** | **89.50** | **88.41** | **89.40** | **90.10** | **90.78** | **92.17** | **93.29** |

We further conduct zero-shot classification tasks on the CheXpert5x200 [15], RSNA [44], and SIIM-ACR [45] datasets. CheXpert5x200 includes five common chest diseases, $Atelectasis$, $Cardiomegaly$, $Consolidation$, $Edema$, and $Pleural\ Effusion$, each with 200 chest X-rays. It is important to note that the CheXpert training set does not include any data from CheXpert5x200, so there is no data leakage issue. The test sets for RSNA and SIIM-ACR are the same as those used in the supervised classification task. All text prompts are provided by a professional radiologist [15]. During testing, we calculated the similarity between image features and text prompt features for all diseases, with the highest similarity indicating the predicted category. As shown in Tab. 2, CLIP [42] performs poorly on medical images due to its training data primarily consisting of natural images. The

Table 2: Comparison results of zero-shot classification tasks with other SOTA models on CheXpert 5x200, RSNA, and SIIM-ACR datasets. The Accuracy (Acc.) and F1-score (F1) metrics are reported. Red and blue denote the best and second-best results.

| Method | CheXpert 5x200 [15] | | RSNA [44] | | SIIM-ACR [45] | |
|---|---|---|---|---|---|---|
| | Acc.↑ | F1↑ | Acc.↑ | F1↑ | Acc.↑ | F1↑ |
| CLIP [42] | 20.10 | 9.12 | 25.03 | 22.07 | 49.39 | 47.98 |
| GLoRIA [15] | 53.30 | 48.99 | 29.15 | 28.54 | 22.57 | 22.57 |
| PRIOR [5] | 34.90 | 30.56 | 76.77 | 51.80 | 50.00 | 33.33 |
| MGCA [53] | 43.60 | 41.37 | 60.83 | 57.77 | 30.03 | 25.45 |
| MedCLIP [55] | 57.50 | 55.97 | 43.09 | 31.01 | 58.40 | 57.85 |
| EGMA(Ours) | 61.30 | 60.38 | 76.97 | 43.49 | 63.62 | 61.46 |

models in rows two to five use encoders pre-trained on medical datasets, and thus, their performance is better than that of CLIP. Interestingly, the GLoRIA [15] and MGCA [53] perform worse than the CLIP model in diagnosing pneumonia on the SIIM-ACR dataset. This indicates that these models are significantly influenced by the data distribution, resulting in poor generalization performance. Conversely, our EGMA achieves the best results in all other metrics, except for the F1-score on the RSNA dataset. This demonstrates that our model, enhanced by eye-gaze data, has learned more generalizable feature relationships between medical images and text, significantly improving its generalization performance.

**Image-text Retrieval** To further validate the alignment capability of our model between visual and textual features, we compare the zero-shot retrieval performance of EGMA with other models on CheXpert 8x200 dataset [59]. Unlike CheXpert5x200 [15], CheXpert8x200 includes eight common chest diseases, $No\ Finding$, $Cardiomegaly$, $Edema$, $Pneumonia$, $Atelectasis$, $Pneumothorax$, $Pleural\ Effusion$, and $Fracture$, each with 200 chest X-rays and five corresponding text prompts. It is worth noting that the prompts for retrieval tasks are different from those for classification tasks in the previous section, but all are written by board-certified radiologists. In the image-to-text retrieval task, we first compute the similarity between the image and all candidate texts, and then rank the retrieved results. Similarly, in the text-to-image task, we compute the similarity between the textual prompts and all images, and rank the retrieval results. We report Precision at Top-1, Top-5, and Top-10, which reflect how many relevant examples are retrieved. As shown in Tab. 3, our model achieves the best results in both retrieve tasks. Our model outperforms the second-best model in the image-to-text and text-to-image retrieval tasks by 3.9%, 5.88%, and 4.33%, and 19.75%, 14.50%, and 12% in terms of P@1, P@5, and P@10 metrics, respectively. This indicates that our model has fully learned the relationship between images and texts, achieving better alignment effects.

Table 3: Comparison results of zero-shot retrieval task with other SOTA models on CheXpert 8x200 dataset. The Precision at Top-1, Top-5, and Top-10 are reported. Red and blue denote the best and second-best results.

| Method | Image-to-text | | | Text-to-image | | |
|---|---|---|---|---|---|---|
| | P@1↑ | P@5↑ | P@10↑ | P@1↑ | P@5↑ | P@10↑ |
| CLIP [42] | 12.75 | 12.48 | 10.03 | 5.00 | 12.50 | 12.50 |
| MedCLIP [55] | 14.50 | 15.98 | 15.86 | 12.50 | 12.50 | 15.00 |
| MGCA [53] | 35.00 | 27.80 | 23.33 | 45.00 | 47.50 | 44.00 |
| GLoRIA [15] | 38.75 | 31.62 | 24.51 | 52.50 | 49.00 | 50.25 |
| ConVIRT [59] | - | - | - | 60.25 | 60.00 | 57.50 |
| EGMA(Ours) | 42.65 | 37.50 | 28.84 | 80.00 | 74.50 | 69.50 |

## 4.2 Ablation Study

To further validate the model's performance, we conducted ablation experiments on the proposed EGF and EGM modules, while also assessing the impact of the proportion of eye-gaze data on the model results. As shown in the upper half of Tab. 4, the first row represents our Baseline model, where we utilize the initialized weights pre-trained on CheXpert [16] and MIMIC-CXR [20] datasets [55]. The

second row "MLCE" indicates that within our EGF module, the EGF loss is not further computed beyond the Eq. 4, instead, only the multi-label cross-entropy (MLCE) loss between the eye-gaze guided similarity matrix and the model's output similarity matrix is calculated. The third row "EGF" utilizes the Eye-gaze Guided Fine-grained loss described in Eq. 6. The fourth row "EGM" indicates that the model is trained solely through the Eye-gaze Guide cross-model Mapping method. Finally, the fifth row presents our proposed EGMA model, which integrates the aforementioned modules guided by eye-gaze data.

Table 4: Comparison results of zero-shot classification ablation experiments on CheXpert 5x200, RSNA, and SIIM-ACR datasets. The Accuracy (Acc.) and F1-score (F1) metrics are reported. Each value in the lower part is the average of three runs. **Red** and blue denote the best and second-best results.

| Method | CheXpert5x200 [16] | | RSNA [44] | | SIIM-ACR [45] | |
|---|---|---|---|---|---|---|
| | Acc.↑ | F1↑ | Acc.↑ | F1↑ | Acc.↑ | F1↑ |
| Baseline | 57.50 | 55.97 | 43.09 | 31.01 | 58.40 | 57.85 |
| MLCE | 60.90 | 59.59 | 47.06 | 33.04 | 27.43 | 22.81 |
| EGF | 60.30 | 58.44 | 53.81 | 35.52 | 63.54 | **65.70** |
| EGM | 59.30 | 57.74 | 54.68 | 35.80 | 52.61 | 47.85 |
| Unified(Ours) | **61.30** | **60.38** | **76.97** | **43.49** | 63.62 | 61.46 |
| 1% Gaze | 58.93±0.06 | 56.62±0.05 | 40.38±0.01 | 29.75±0.01 | 57.90±0.21 | 57.37±0.24 |
| 5% Gaze | 58.93±.006 | 56.69±0.06 | 53.00±0.05 | 35.13±0.01 | 59.20±0.11 | 58.51±0.10 |
| 10% Gaze | 59.30±0.01 | 57.82±0.01 | 53.78±0.01 | 35.37±.001 | 58.27±0.11 | 57.71±0.12 |
| 50% Gaze | **59.55±0.07** | **58.84±0.01** | **58.54±0.07** | **37.01±0.20** | **61.41±0.31** | **58.88±0.22** |

In Tab. 4, it can be observed that the method using only gaze-guided MLCE loss significantly improves performance compared to the baseline on CheXpert 5x200 dataset, with a slight improvement on RSNA but a severe decline on SIIM-ACR dataset. However, models using EGF or EGM show significant improvements on SIIM-ACR. This indicates that while MLCE improves performance on some datasets, it simultaneously reduces the model's generalization ability. Thus, relying solely on simple loss for similarity matrix is insufficient. In this work, by combining eye-gaze guided image-text relationships with fine-grained feature alignment (EGF), although the model's performance slightly decreases on CheXpert 5x200, its overall generalization improves. Similarly, to enhance the model's multi-modal alignment ability, introducing eye-gaze guided cross-modal mapping results in improved performance and generalization, with EGM achieving optimal performance on RSNA dataset. Finally, when optimizing both fine-grained alignment and cross-modal alignment using eye-gaze, the model achieves dominant performance on all three datasets, demonstrating further enhancement in generalization.

Numerous studies [24, 54, 35, 36] have demonstrated that training models using eye-gaze data can achieve comparable performance to models trained with fine-grained manual annotations. Meanwhile, the cost of collecting fine-grained manual annotations is significantly higher than that of collecting eye-gaze data. Therefore, incorporating eye-gaze into pre-training tasks is a feasible approach to enhancing model performance. To further validate the efficiency of our model using eye-gaze data, we conduct ablation experiments on the proportion of it in the training set. Our training dataset, MIMIC-EYE, consists of a total of 3695 samples. We perform ablation experiments using 1%, 5%, 10%, and 50% of the eye-gaze data, resulting in 37, 185, 370, and 1848 samples with prior information from radiologists, respectively. We repeat each experiment three times to eliminate the bias caused by random sampling, and report the average results. As shown in the lower part of Tab. 4, the model's performance on the CheXpert 5x200 dataset improved when trained with 1% of eye-gaze data. However, due to the limited data volume, the model's performance on other datasets is inferior to the baseline. When increasing the eye-gaze data to 5%, the model shows significant improvements on all three datasets. With the continuous increase in eye-gaze data, the performance of the model also improves. Therefore, even with a small amount of eye-gaze data (185 samples), our framework can effectively guide the model's multi-modal processing capability, ensuring performance enhancement. This further illustrates the applicability of our model and its low training cost characteristics.

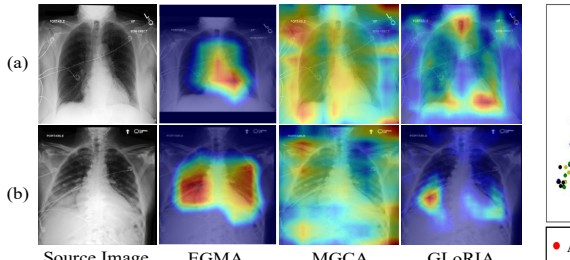

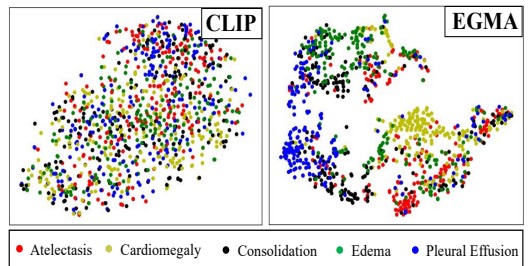

| Source Image | EGMA | MGCA | GLoRIA |

Figure 3: Results of cross-modality attention maps visualization. Related text content: (a) "*heart size borderline enlarged*"; (b) "*increased bibasilar opacities are the combination of increased bilateral pleural effusions and bibasilar atelectasis*".

Figure 4: t-SNE visualization on CheXpert 5x200 dataset by CLIP and our EGMA. The figures display points of different colors representing various ground truth disease types and their cluster assignments. The color-coded points illustrate the clustering results of each algorithm.

## 4.3 Visualization

To better demonstrate the correspondence learned by the EGMA framework between text and radiographic images, we conducted a cross-modality attention maps visualization in Fig. 3. Guided by eye-gaze data, the EGMA framework clearly outperforms other state-of-the-art methods in the field in accurately localizing disease regions. In Fig. 4, we visualize the feature representations of CLIP [42] and our EGMA model on images of CheXpert 5x200 dataset using the t-SNE [50]. It can be observed that our model exhibits better clustering representation. The CLIP model, which was not trained on medical data, is unable to effectively differentiate these diseases. More results of t-SNE visualization can be referred to the Appendix E, clustering performance of other SOTA methods [15, 53] also inferior to our EGMA.

## 5 Discussion and Conclusion

In this work, we reveal the significant role of radiologists' eye-gaze data in multi-modal alignment and propose an **E**ye-gaze **G**uided **M**ulti-modal **A**lignment framework called EGMA. In fact, the EGF and EGM modules in our EGMA framework are similar to the diagnostic process of radiologists. Many studies [39, 46, 30, 8] have pointed out that radiologists perform a global search first and then conduct a detailed examination of the local areas when suspicious lesions are found. Firstly, the EGF module of EGMA corresponds to the global search by radiologists, aligning the local image patch features with the individual sentence features. Secondly, EGM uses the key values from the local similarity matrix computed in the first step as weights, focusing the alignment on certain important image patches and texts, akin to the detailed observation stage after identifying suspicious lesions. Thus, by mimicking the real diagnostic behavior of radiologists, our EGMA further enhances the model's multimodal processing capability and diagnostic accuracy. We evaluate EGMA's zero-shot capabilities and fine-tuned performances on multiple datasets and observe significant improvement in classification and retrieval tasks. Additionally, we investigate the impact of eye-gaze data scale on performance, finding that even small amounts of eye-gaze data can enhance the model's multi-modal alignment capabilities during pre-training. Overall, our EGMA framework explores the feasibility of incorporating eye-gaze data from radiologists to assist in multi-modal feature alignment during model training, laying the foundation for the application of eye-gaze data in the medical multi-modal domain.

**Limitations and Discussion** Our work only compared state-of-the-art methods in classification and retrieval tasks, without conducting downstream tasks such as lesion localization or segmentation. Additionally, our model heavily relies on multi-modal datasets like MIMIC-EYE [14], which can simultaneously collect eye-gaze data, medical images, and diagnostic text. The scenarios for collecting these data are also a significant consideration. For instance, in clinical ultrasound diagnosis [38], radiologists often use both hands to operate the equipment and verbally communicate their diagnostic information to an assistant. In this context, it is convenient to simultaneously record ultrasound images, eye-gaze data, and audio. In contrast, during chest X-ray diagnosis in MIMIC-EYE, radiologists typically record diagnostic information directly in text form rather than verbally. Fortunately, some

recent efforts [25, 35] are focusing on how to naturally collect multi-modal data of radiologists during diagnosing. They have designed more flexible collection systems that better accommodate the routine work of radiologists, which is crucial for the widespread adoption of collecting multi-modal diagnostic data such as eye-gaze information.

**Potential Impacts** Although the eye-gaze data we used is publicly available and we have permission to use it, some studies [27, 23] have indicated that private information such as gender, age, and mental state of observers can be extracted from eye-gaze data. Therefore, privacy concerns have always been a focal point in using eye-gaze data. To address this, we recommend using de-identification methods to filter eye-gaze data or releasing the data in the form of heatmaps rather than the raw data.

**Future Work** In the future, we will continue to optimize these proposed collection systems [27, 23] and explore the guidance role of eye-gaze data between images and handwritten diagnostic reports to accelerate their application in real medical diagnostic scenarios. This will provide a research foundation to alleviate data annotation pressure and enhance model interpretability. Additionally, we will continue to analysis the eye-gaze features, such as temporal features, and further optimize their role in multi-modal feature alignment. We believe this work can serve as a valuable reference for the application of eye-gaze data in multi-modal frameworks and promote its development in the field of medical multi-modality. Moreover, thanks to the inherent flexibility of the EGMA model, it is well-suited for multi-modal alignment tasks involving natural images, such as in human-robot interaction and control as well as for education/training purposes. Therefore, we believe this is a promising direction for future expansion.

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

# A  Appendix

This appendix is organized as follows. In Sec. B, we provide more experimental settings, including training parameters, detailed parameters of image and text encoders. In Sec. C, we introduce the detailed information of the datasets used in this work. In Sec. D, we provide more details of multi-modal data processing of MIMIC-EYE [14] dataset. In Sec. E, we provide additional visualization results of feature representation. In Sec. F, we provide additional experiments of zero-shot classification task after continue pre-training using the backbones of other SOTA models in our EGMA framework.

# B  Experimental Details

## B.1  Image/Text Encoder

In this study, we use SwinTransformer [33] as the image encoder, BioClinicalBERT [1] as the text encoder. Specifically, we use a 4 stages SwinTransformer, including 2, 2, 6, and 2 SwinTransformer blocks. Other parameters are: patch size 4; window size 7. And we use a 6 layers BioClinicalBERT with 12 attention heads. In our EGMA framework, we add a linear projection layer after both the image encoder and text encoder to map the embeddings' dimension to 512, and we use a learnable temperature $\tau$ in contrastive loss calculation initialized on 0.07.

## B.2  Training Settings

**Pre-training Settings** In the pre-training process, we utilize the following image augmentations to the chest X-ray images: scale to images to $224 \times 224$; color jittering with brightness and contrast ratios from [0.8, 1.2]; randomly change the contrast($probability = 0.5$). And we train our model with 50 epochs with an initial learning rate $1 \times 10^{-6}$ and weight decay $1 \times 10^{-4}$ and 10 epochs of warm-up.

**Fine-tuning Settings** In the supervised classification experiments, we adopt the linear classification settings [15], where the pre-trained image encoder is frozen, and only a randomly initialized linear classification head is trained. We choose the same image augmentations to the above pre-training settings. And we fine-tune our model with 30 epochs with an initial learning rate $5 \times 10^{-7}$ and weight decay $1 \times 10^{-4}$ and 6 epochs of warm-up. And all our training tasks are completed on four RTX 3090 GPUs.

# C  Dataset Descriptions

## C.1  MIMIC-EYE

The MIMIC-EYE [14] dataset includes a comprehensive range of patient information, including medical images and reports, clinical data, patient's hospital journey, and eye-tracking data and audio of radiologists during diagnosis. The dataset comprises a total of 3689 images from the MIMIC-IV v1.0 dataset [18], each accompanied by transcripts text from audio and eye-tracking data of radiologists. In this work, we use this dataset as our training set.

## C.2  CheXpert

CheXpert [16] is a large-scale public dataset for chest radiograph interpretation, developed by a team from Stanford University. The dataset comprises 224,316 chest radiographic images involving 65,240 patients, annotated for the presence of 14 common chest radiographic findings [13]. These annotations are categorized into three types: positive, negative, or uncertain. In our study, we follow [15] and [59], using two subsets of this dataset, namely CheXpert 5x200 and CheXpert 8x200, for our zero-shot classification and zero-shot retrieval testing tasks. The CheXpert 5x200 dataset [15] comprises five common chest diseases, $Atelectasis$, $Cardiomegaly$, $Consolidation$, $Edema$, and $Pleural\ Effusion$, each with 200 chest X-rays. In [15], a radiologist provided possible sub-types, severities, and locations for these five diseases. As depicted in Tab. 5, all combinations of these three types of information form the text queries for CheXpert 5x200 dataset. In the zero-shot classification

Table 5: Examples of possible sub-types, severities, and locations provided by the radiologist in CheXpert 5x200 dataset.

|  | Atelectasis | Consolidation | Pleural Effusion |
|---|---|---|---|
| severity | mild
minimal | increased
improved
apperance of | small
stable |
| subtype | subsegmental atelectasis
linear atelectasis
trace atelectasis
bibasilar atelectasis
retrocardiac atelectasis
bandlike atelectasis | bilateral consolidation
reticular consolidation
patchy consolidation
airspace consolidation
partial consolidation | bilateral pleural effusion
subpulmonic pleural effusion
bilateral pleural effusion |
| location | at the mid lung zone
at the upper lung zone
at the right lung zone
at the left lung zone
at the lung bases | at the lower lung zone
at the upper lung zone
at the left lower lobe
at the right lower lobe
at the left upper lobe | left
right
tiny |

task, image embeddings are compared with the embeddings of these text queries, and the class with the highest similarity is assigned as the predicted classification for the image. The CheXpert 8x200 dataset [59] comprises eight categories, $NoFinding$, $Cardiomegaly$, $Edema$, $Pneumonia$, $Atelectasis$, $Pneumothorax$, $Pleural\ Effusion$, and $Fracture$, each with 200 images. In [59], a radiologist expert was also invited to compose five expert queries for each category, used for image-text retrieval tasks. Specific queries are detailed in Tab. 6.

### C.3  RSNA

The RSNA Pneumonia Detection Dataset [44], encompasses a comprehensive set of medical imaging data types, including X-rays, CT (Computed Tomography), and MRI (Magnetic Resonance Imaging) images. In this work, we utilized the stage 2 version of this dataset, comprising 29,700 chest X-ray images categorized into $normal$ and $pneumonia$ positive category. Following [15], we allocated 15% of this dataset for our zero-shot classification testing set. And we utilize the text queries from the "no finding" and "Pneumonia" categories in the CheXpert 8x200 dataset as the text queries for zero-shot classification in this data.

### C.4  SIIM-ACR

The SIIM-ACR [45] dataset is a chest dataset used for $pneumothorax$ classification and segmentation. It consists of 2379 images with pneumothorax and 8300 images without pneumothorax. In this study, we utilized a subset of the dataset filtered by Saab et al. [43] as the test data to evaluate the zero-shot classification performance of the model for pneumothorax disease. And we utilize the text queries from the "no finding" and "Pneumothorax" categories in the CheXpert 8x200 dataset as the text queries for zero-shot classification in this data.

## D  Details of Multi-modal Data Processing

As illustrated in Fig. 5, the presentation of multi-modal data in the MIMIC-EYE [14] dataset includes radiologists' audio, text transcript, eye-gaze data, and image. Since each modality is synchronized, the audio data is aligned with the eye-gaze data in time. By segmenting the audio based on the time before and after the pronunciation of each word, we can align the transcripts with the audio, thereby aligning word-level text with eye-gaze data. Subsequently, we generate attention heatmap based on eye-gaze data and images to represent the image regions the radiologist focuses on. Through the aforementioned data processing steps, we achieve precise alignment between word-level text and image regions. It is noteworthy that due to the rapid speech rate of radiologists, there may be no

Table 6: Examples of text queries for different categories in the CheXpert 8x200 dataset.

| Categories | Text Query |
| --- | --- |
| No Finding | The lungs are clear.
No abnormalities are present.
The chest is normal.
No clinically significant radiographic abormalities.
No radiographically visible abnormalities in the chest. |
| Cardiomegaly | The heart is mildly enlarged.
Cardiomegaly is present.
The heart shadow is enlarged.
The cardiac silhouette is enlarged.
Cardiac enlargement is seen. |
| Edema | Mild interstitial pulmonary edema is present.
The presence of hazy opacity suggests interstitial pulmonary edema.
Moderate alveolar edema is present.
Mild diffuse opacity likely represents pulmonary edema.
Cardiogenic edema likely is present. |
| Pneumonia | A consolidation at the base likely represents pneumonia.
Pneumonia is present.
The presence of air bronchograms suggest pneumonia.
A fluffy opacity suggests pneumonia.
A pulmonary opacity with ill defined borders likely represents pneumonia. |
| Atelectasis | Platelike opacity likely represents atelectasis.
Geometric opacity likely represents atelectasis.
Atelectasis is present.
Basilar opacity and volume loss is likely due to atelectasis.
Patchy atelectasis is seen. |
| Pneumothorax | An apical pneumothorax is present.
A basilar pneumothorax is seen.
A medial pneumothorax is present adjacent to the heart.
A lateral pleural line suggests pneumothorax.
Pleural air is present. |
| Pleural Effusion | A pleural effusion is present.
Blunting of the costophrenic angles represents pleural effusions.
Trace pleural fluid is present.
The pleural space is partially filled with fluid.
Layering pleural effusions are present. |
| Fracture | An angulated fracture is present.
An oblique radiolucent line suggests a fracture.
A cortical step off indicates the presence of a fracture.
A communuted displaced fracture is present.
A fracture is present. |

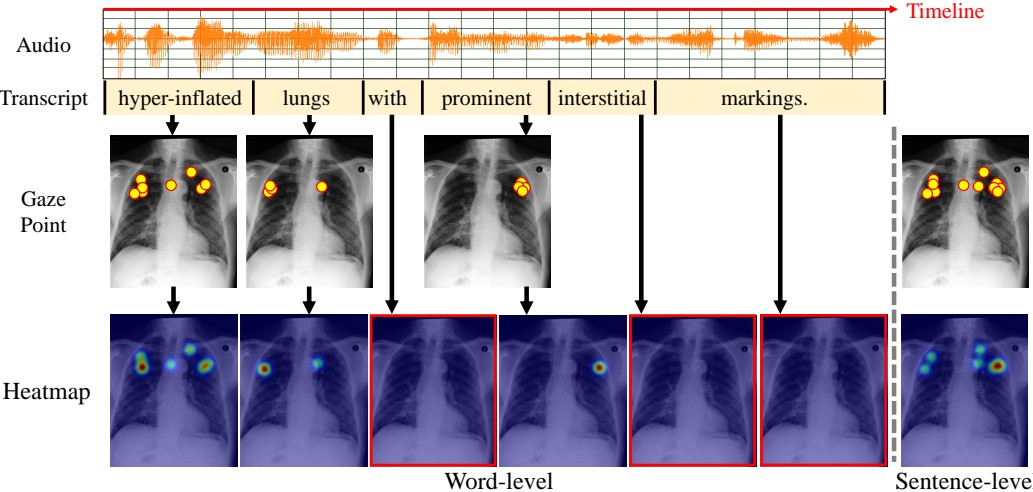

Figure 5: The generation methods for heatmap at both word-level and sentence-level.

available eye-gaze data within the time interval corresponding to a single word. In Fig. 5, the word "with" in the transcript has no corresponding gaze data. Another common and unavoidable issue is the loss of eye-gaze data caused by blinking and intense head movement of radiologist, as seen in the last two words of the transcript. Due to these technical challenges, achieving perfect pairing between words and image regions is difficult. However, as shown in the right side of Fig. 5, adjusting the text to the sentence level largely mitigates the issue of missing word-level heatmap (Heatmap with red edge), and the semantic information of the entire sentence also encompasses the information of each word. Therefore, in this work, we process text features at the sentence level.

During the pre-training of EGMA, the size of the input heatmap is determined by the number of patches in the image. For example, after the image is processed by the image encoder, the size of image embedding is $196 \times 768$, where 196 represents the number of image patches. Therefore, we resize the heatmap directly to $14 \times 14$ to match the image embedding and further process it into the Gaze-guided Similarity and Gaze-guided Label mentioned in the main manuscript.

### D.1 Eye-gaze Data Denoising

In fact, noise and errors in eye-gaze data are as common as noise in images, and these can all affect the model's final performance. In this work, the errors in eye-gaze data primarily stem from two factors: involuntary saccades and microsaccades of the radiologists' eyes [9], and subjective fixation errors [3].

Human eye movements can be categorized into two main types: saccades and fixations [9]. Saccades are the rapid movements between different areas of the visual field and are generally considered noise data that do not involve cognitive processes. Fixations, on the other hand, are brief pauses of the eyes on a small area and are regarded as the primary cognitive behavior. Additionally, due to the structure of the eye, fixations are accompanied by microsaccades, causing the fixation point to drift within the fixation area. Thus, microsaccades are also a major source of noise in eye-gaze data. Fortunately, eye-tracking technology has evolved significantly over the past century, and noise reduction techniques for eye-gaze data have become highly advanced. Currently, all commercial eye-tracking devices come with preprocessing software that uses adaptive methods (e.g., [41]) to filter out noise and output fixations, greatly facilitating the use of eye-tracking technology. Fig. 6 shows an example of MIMIC-EYE data used in our work. All gaze data is denoised by the filter operation, ensuring that noise from blinks, saccades, and microsaccades of the radiologists has been eliminated.

Additionally, due to variations in radiologists' expertise and cognitive levels, fixation data may contain errors, such as the reviewer mentioned, "occasionally shift to locations not exactly on the objects."

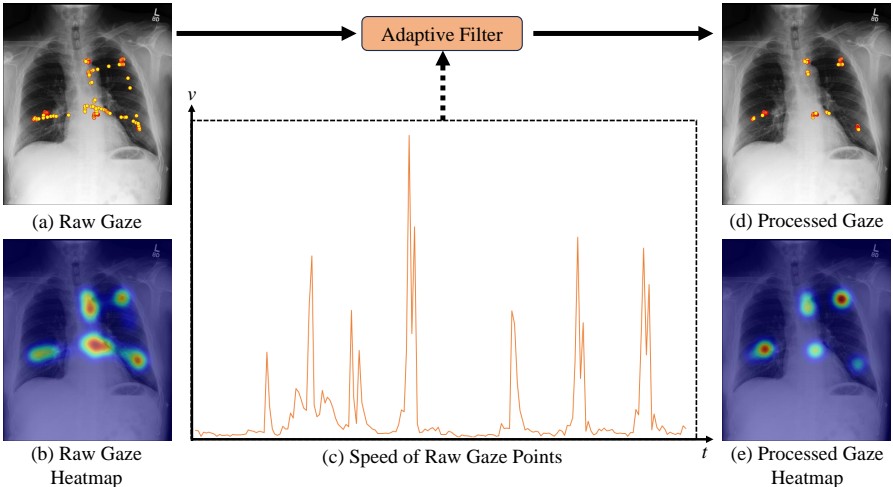

(a) Raw Gaze

(d) Processed Gaze

(b) Raw Gaze Heatmap

(c) Speed of Raw Gaze Points

(e) Processed Gaze Heatmap

Figure 6: Attention heatmap (b) generated from raw gaze data (a) is susceptible to noise. The adaptive filter employed in the preprocessing step of this work removes noisy data (saccades and microsaccades) based on characteristics such as the speed of gaze points (c), resulting in more accurate fixation data (d) and heatmap (e).

For the minor shift error of fixations, where the duration of incorrect fixations is small compared to the overall duration, their values in the attention heatmap are low, thus having minimal impact on the mlce loss utilized in our work (Eq. 4 in main paper). In our EGMA, EGF module (Sec. 3.2 in main paper) selects the highest similarity value of local features before averaging, ensuring that these minor errors do not affect the model. Additionally, in the EGM module (Sec. 3.3 in main paper), the sparse operation for alignment weight also filters out minor errors in the heatmap. For the significant fixation errors, MIMIC-EYE data has been cleaned to address this issue at the first time. Specifically, during data collection, radiologists were required to provide a diagnostic label. These data were then reviewed by professionals to exclude inconsistent diagnoses and fixation errors. Moreover, the MIMIC-EYE dataset includes eye-gaze data from six radiologists. Even if one or two radiologists' fixations for a specific diagnostic text are incorrect, the correct fixation data from the other radiologists under the same semantic context can mitigate the adverse effects. Fig. 7 shows an example of this compensation.

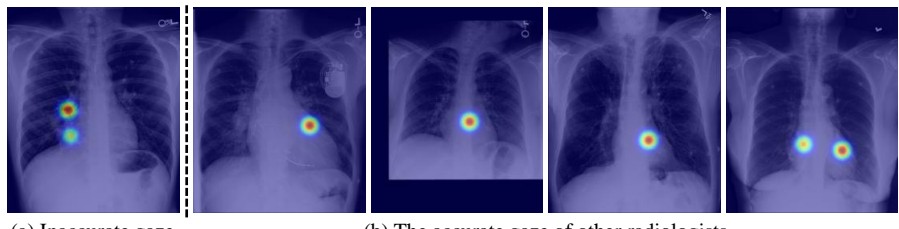

(a) Inaccurate gaze

(b) The accurate gaze of other radiologists

Figure 7: Inaccurate eye-gaze data of one radiologist (a) in the **heart region** and several correct eye-gaze data (b) of other radiologists in the same region that compensate for this error, which are included in the dataset used in this work.

In summary, EGMA was designed with considerations for the noise and errors inherent in eye-gaze data from the beginning. And effectively identifying and mitigating errors in radiologists' diagnostic behavior during model training will be a focus of our future work.

## D.2    Eye-gaze Data Properties

Numerous studies have investigated and proven the close relationship between radiologists' gaze behavior, image content, and diagnostic results. For example, studies [40, 52, 11, 49, 48] have found

that radiologists fixate more on diseased areas than on healthy ones. Moreover, they discovered that novice radiologists, due to their lack of experience, repeatedly look at diseased areas, resulting in even more fixations in these areas than those of expert radiologists. Fig. 8 shows some comparison cases. Additionally, the abnormal regions with more fixations have higher weights in the attention heatmap. Our EGMA model can leverage these weights, along with image content and text, to learn better disease diagnosis capabilities and feature representations.

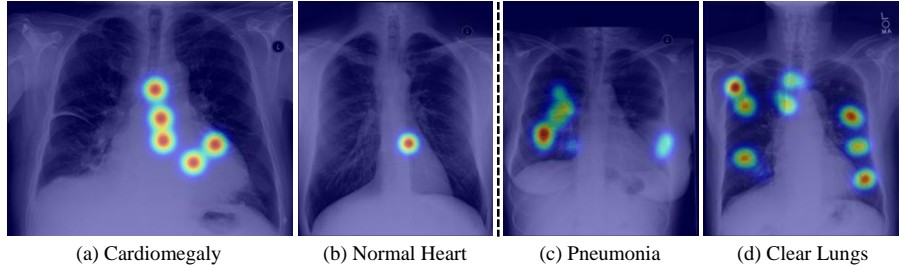

(a) Cardiomegaly      (b) Normal Heart      (c) Pneumonia      (d) Clear Lungs

Figure 8: Comparison of eye-gaze data in normal and abnormal cases. For the heart region, there are more fixations on disease area (a) compared to normal heart (b). For the lung region, fixations on disease area (c) are more concentrated, whereas fixations on normal lungs are more dispersed.

## E    Additional Visualization Results

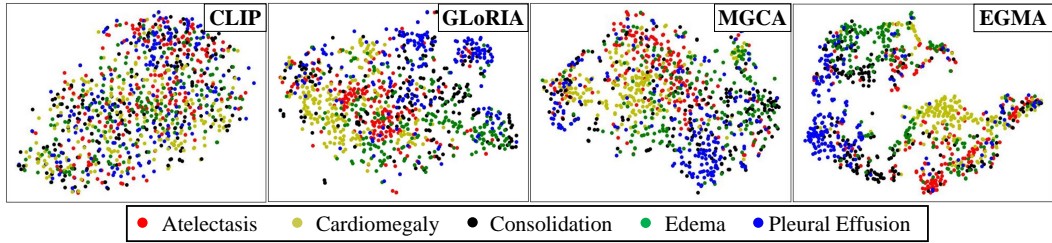

● Atelectasis    ● Cardiomegaly    ● Consolidation    ● Edema    ● Pleural Effusion

Figure 9: Visualization of feature representation of CheXpert 5x200 dataset by CLIP, GLoRIA, MGCA, and our EGMA.

In Fig. 9, we visualize the feature representations of CLIP [42], GLoRIA [15], MGCA [53], and our EGMA model on images of CheXpert 5x200 dataset using the t-SNE [50]. It can be observed that our model exhibits better clustering representation. The CLIP model, which was not trained on medical data, is unable to effectively differentiate these diseases. Additionally, while the representation capability of GLoRIA and MGCA has improved noticeably, their clustering performance still inferior to our EGMA.

## F    Additional Analysis Results

To further validate the generality of our framework, we utilize the encoders and pre-training weights provided by CLIP [42], GLoRIA [15], and MGCA [53] in our EGMA framework. Subsequently, we continue training on the MIMIC-EYE [14] dataset and present the zero-shot classification results on the CheXpert 5x200 [16], RSNA [44], and SIIM-ACR [45] datasets in Tab. 7.

We present accuracy Accuracy and F1 score metrics on three datasets. The values in parentheses indicate the improvement over the baseline metrics (as shown in Tab. 2). It can be observed that, all models show improvement after training with our EGMA framework, except for the decrease in metrics for the trained MGCA model on the RSNA dataset. For MGCA, when tested with its provided pre-trained weights, it performs the best F1-score on the RSNA dataset (as shown in Tab. 2), but after training with the EGMA framework, its performance improves on CheXpert 5x200 and SIIM but decreases on RSNA dataset. This may reflect that the features extracted by MGCA on

Table 7: Comparison results of zero-shot classification after continue pre-training using the backbones of other SOTA models in our EGMA framework. **Red** and blue denote the best and second-best results. The values in (parentheses) represents the improvement over the baseline metrics in Table 1 of main manuscript.

| Method | CheXpert 5x200 [16] | | RSNA [44] | | SIIM-ACR [45] | |
|---|---|---|---|---|---|---|
| | Acc.↑ | F1↑ | Acc.↑ | F1↑ | Acc.↑ | F1↑ |
| CLIP [42] | 20.30(0.2) | 10.73(1.61) | 34.04(9.01) | 33.68(11.61) | 50.19(0.8) | 49.03(1.05) |
| GLoRIA [15] | 54.40(1.1) | 49.31(0.32) | 49.11(19.96) | 38.82(7.81) | 31.07(8.5) | 31.10(8.53) |
| MGCA [53] | 50.20(6.6) | 48.29(6.92) | 57.08(-3.75) | 40.40(-17.37) | 32.65(2.62) | 27.78(2.33) |
| EGMA(Ours) | 61.30 | 60.38 | 76.97 | 43.49 | 63.62 | 61.46 |

RSNA dataset are not truly disease-related features but rather shortcut features, indicating that the high baseline metrics were based on easily distinguishable shortcut features. Furthermore, after training with EGMA, the performance of other models significantly improves on all three datasets.

