# OpenReview forum: "Eye-gaze Guided Multi-modal Alignment for Medical Representation Learning"
_NeurIPS.cc/2024/Conference — NeurIPS 2024 poster_

### Official Review · Reviewer_n21y · 2024-06-22

**Soundness:** 3
**Presentation:** 3
**Contribution:** 3
**Rating:** 5
**Confidence:** 2

**Summary:**

This paper introduces the Eye-gaze Guided Multi-modal Alignment (EGMA) framework, which leverages radiologists' eye-gaze data to enhance the alignment of medical visual and textual features. By using synchronously collected eye-gaze data during diagnostic evaluations, EGMA improves generalization and achieves state-of-the-art performance in image classification and image-text retrieval tasks across four medical datasets. The study also investigates the impact of varying amounts of eye-gaze data on model performance, demonstrating the feasibility and utility of integrating this auxiliary data into multi-modal alignment frameworks.

**Strengths:**

Innovative Use of Eye-Gaze Data:
The introduction of the Eye-gaze Guided Multi-modal Alignment (EGMA) framework is a novel approach that leverages eye-gaze data from radiologists to improve the alignment of visual and textual features in medical data. This innovative use of auxiliary data opens new avenues for enhancing model performance and provides a unique perspective on incorporating human cognitive processes into machine learning models.
Empirical Validation and Generalization:
The EGMA framework is rigorously evaluated across four different medical datasets, demonstrating its ability to achieve state-of-the-art performance in image classification and image-text retrieval tasks. This comprehensive empirical validation underscores the robustness and effectiveness of the proposed method, highlighting its potential for broader application in various medical contexts.
Exploration of Eye-Gaze Data Utility:
The paper not only introduces a novel framework but also explores the impact of varying amounts of eye-gaze data on model performance. This detailed analysis provides valuable insights into the feasibility and utility of integrating eye-gaze data into multi-modal alignment frameworks, offering practical guidelines for future research and application in the field.

**Weaknesses:**

Dependency on Eye-Gaze Data Collection:
Issue: The reliance on eye-gaze data from radiologists introduces a significant dependency that may not be feasible for all institutions due to the need for specialized equipment and the additional effort required to collect this data. Especially when the eye-gaze prior is not accurate.
Impact: This dependency could limit the scalability and widespread adoption of the EGMA framework, particularly in resource-constrained settings or where the collection of eye-gaze data is not practical.

**Questions:**

Impact of Eye-Gaze Data Quality and Quantity:
Question: How does the quality and quantity of eye-gaze data affect the performance of the EGMA framework? Are there specific thresholds or optimal amounts of eye-gaze data required to achieve the best results?
Context: Detailed insights into the relationship between eye-gaze data quality/quantity and model performance would help in optimizing the framework and understanding its practical requirements.

---

> ### Author Rebuttal · Authors · 2024-08-07
>
> **W: "The reliance on eye-gaze data from...where the collection of eye-gaze data is not practical."**
>
> We thank the reviewer for this great comment. In fact, when designing experiments for zero-shot classification and zero-shot image-text retrieval, EGMA took this issue into account. Experimental validation has shown that after multimodal training on datasets with eye-gaze data, the model performs well on other datasets without eye-gaze data (Tab. 1\&2\&3 in main paper). Furthermore, with advancements in eye-gaze data collection systems [1-4], acquiring multimodal data with eye-gaze has become increasingly feasible. Compared to having radiologists spend additional time and effort annotating refined labels, eye-tracking allows for data collection during their routine diagnostic work, minimizing disruption and reducing annotation burdens. Additionally, some studies have demonstrated [3,5,6] that eye-gaze data can achieve performance comparable to refined annotations. Therefore, we propose initially pre-training on scenarios where eye-gaze data is more easily collected and then fine-tuning on more challenging datasets. This approach is similar to current mainstream multimodal pre-training models, where a robust model is trained on a broadly available dataset and then fine-tuned on other datasets to perform downstream tasks. The performance of EGMA in this work supports the feasibility of this approach.
>
>
> **Q1: "How does the quality and quantity of eye-gaze data affect the performance of the EGMA framework?"**
>
> We thank the reviewer for this great question. The quality of eye-gaze data is crucial for model performance. In this work, we first filter out interfering data such as blinks and saccades by using only the fixation data from the eye-tracking recordings. We then generate a 2D attention heatmap from these fixation points to further mitigate the influence of interfering data. Figure R1 in the attached PDF shows an example of denoising in MIMIC-EYE dataset. Additionally, in the design of the EGMA framework, we use the EGM module to further filter out potentially problematic regions. For catastrophic errors could happened, the MIMIC-EYE dataset provides a robust solution by involving multiple expert radiologists in data collection. This approach ensures overall data quality and allows for the compensation of errors made by one or two radiologists with correct data from other radiologists. Figure R2 in the attached PDF shows an example of this compensation.
>
> Regarding the issue of the quantity of eye-gaze data, we conducted extensive comparisons in the ablation study of this work (Tab. 4 in main paper). Specifically, we evaluated the model performance with 1\%, 5\%, 10\%, and 50\% of the eye-gaze data. We found that with a small amount of data, such as 1\% (approximately 37 samples), the model's performance improvement was limited, which is attributed to insufficient auxiliary information. As the proportion of eye-gaze data increased, the performance of EGMA improved progressively across the three datasets. This aligns with the expectation that providing more supervised information allows the model to learn better feature representations and consequently enhance performance on the datasets.
>
> **Q2: "Are there specific thresholds or optimal amounts of eye-gaze data required to achieve the best results?"**
>
> We thank the reviewer for this great question. For our EGMA framework, having more eye-gaze data is certainly beneficial, similar to other types of annotations. Additionally, EGMA is designed to handle situations where eye-gaze data may be limited, as it supports training with both eye-gaze and non-eye-gaze data. In our ablation experiments (Tab. 4 in main paper), we also investigated the impact of the proportion of eye-gaze data on model performance. Our goal was to demonstrate that even with limited eye-gaze data (e.g., using only 185 samples), EGMA's performance can still improve steadily, highlighting the framework's flexibility and effectiveness.
>
> References:
>
> [1] Stember, J.N., et al. (2020). Integrating eye tracking and speech recognition accurately annotates MR brain images for deep learning: proof of principle.
>
> [2] Khosravan, N., et al. (2019). A collaborative computer aided diagnosis (C-CAD) system with eye-tracking, sparse attentional model, and deep learning.
>
> [3] Ma, C., et al. (2023). Eye-gaze-guided vision transformer for rectifying shortcut learning.
>
> [4] Men, Q., et al. (2023). Gaze-probe joint guidance with multi-task learning in obstetric ultrasound scanning.
>
> [5] Wang, S., et al. (2022). Follow my eye: Using gaze to supervise computer-aided diagnosis.
>
> [6] Ma, C., et al. (2023). Rectify vit shortcut learning by visual saliency.

---

### Official Review · Reviewer_V2ie · 2024-07-07

**Soundness:** 4
**Presentation:** 3
**Contribution:** 4
**Rating:** 7
**Confidence:** 4

**Summary:**

The paper proposes EGMA, a novel framework for medical multi-modal alignment integrating eye-gaze data into vision-language pre-training. EGMA outperforms existing methods in image classification and image-text retrieval tasks, demonstrating significant advancements and improved feature representation with even minimal eye-gaze data.

**Strengths:**

Innovative approach that diverges from traditional reliance on annotated datasets, providing a fresh perspective on multi-modal learning in medical contexts.

Robust experimental design that includes comparisons with state-of-the-art methods, highlighting the efficacy of the proposed framework.

Comprehensive visualizations and tables that effectively communicate the results and support the paper's claims.

Provides a scalable solution that shows strong generalization across different datasets, indicating its broader applicability in various medical settings.

**Weaknesses:**

The paper primarily focuses on classification and retrieval tasks. Including additional evaluations, such as lesion localization or segmentation tasks, could provide a more comprehensive assessment of the framework’s effectiveness.

The use of eye-gaze data raises potential privacy issues, as such data can inadvertently reveal sensitive information about the observers. Addressing these concerns through robust de-identification methods would strengthen the ethical considerations of the work.

Although the paper includes ablation studies, more detailed analysis on the impact of different components of the model, such as varying the amount of eye-gaze data or different types of medical images, could provide deeper insights into the robustness and adaptability of the proposed framework.

The model relies on multi-modal datasets like MIMIC-EYE, which simultaneously collect eye-gaze data, medical images, and diagnostic text. This dependency might limit the generalizability of the approach to other datasets that lack such rich multi-modal annotations.

**Questions:**

The paper mentions that even a small amount of eye-gaze data can enhance model performance. Can you provide more quantitative results or analysis on how performance scales with different amounts of eye-gaze data?

Have you considered applying the EGMA framework to other types of medical imaging modalities (e.g., MRI, CT scans) or diagnostic tasks? If so, what challenges or modifications would be necessary?

**Limitations:**

The authors discuss privacy concerns associated with using eye-gaze data, suggesting the use of de-identification methods or releasing data in the form of heatmaps to mitigate these issues.

---

> ### Author Rebuttal · Authors · 2024-08-07
>
> **W1:** We greatly appreciate the reviewer's suggestion. Supervision information used for localization and segmentation tasks, such as bounding boxes and masks, is stronger than the labels used for classification tasks. Additionally, the amount of refined manual annotation required for localization and segmentation is relatively limited. Therefore, we believe that if EGMA performs well on classification tasks with relatively weak supervision and across multiple classification datasets, it is likely to also excel in localization and segmentation tasks. Furthermore, the visualization results in our work already provide preliminary evidence of EGMA's localization capability. Finally, thank the reviewer again for this suggestion. Conducting additional validation experiments is indeed a promising direction for our future work.
>
> **W2:** We thank the reviewer for this great comment. Privacy concerns in eye-gaze data are often overlooked by researchers. Studies [1,2] have shown that eye-gaze data can reveal private information such as the collector's personality traits, age, gender, and mental state, making privacy protection crucial. One approach [3] involves adding noise to the original eye-gaze data to obscure some private information, but this can compromise the quality of the data. Another commonly used method [4], which is employed in EGMA, is to convert the original eye-gaze data into a two-dimensional attention heatmap. This heatmap retains rich visual information while mitigating the risk of privacy leaks compared to raw eye-gaze data. Therefore, the EGMA framework protects the privacy of the participants during the data preprocessing phase.
>
> **W4:** We thank the reviewer for this great comment. In fact, when designing experiments for zero-shot classification and zero-shot image-text retrieval, EGMA took this issue into account. Experimental results show that after multimodal training on datasets with eye-gaze data, the model performs well on other datasets without eye-gaze data. Furthermore, with the advancement of eye-gaze data collection systems [5-8], collecting multimodal data with eye-gaze has become increasingly feasible. Therefore, we propose a strategy where pre-training is conducted in scenarios where eye-gaze data is more easily collected, followed by fine-tuning on more challenging scenarios. This approach is similar to mainstream multimodal pre-training models, where a strong pre-trained model is first trained on a widely available dataset and then fine-tuned on other datasets for downstream tasks. The performance of EGMA in this work supports the feasibility of this approach.
>
> **W3\&Q1:** We thank the reviewer for this great question. In fact, we have discussed this issue in the ablation study of our paper (Tab. 4 in main paper). Given the current scarcity of eye-gaze data, EGMA was designed to support training with both eye-gaze and non-eye-gaze data within the same batch. In Tab. 4, we conducted comparative experiments with four different proportions of eye-gaze data: 1\%, 5\%, 10\%, and 50\%, corresponding to approximately 37, 185, 370, and 1848 instances with eye-gaze data in the training set, respectively. It can be observed that with 1\% of eye-gaze data, the model shows only slight improvement on the CheXpert5x200 dataset due to limited auxiliary information. As the proportion of eye-gaze data increases, the performance of EGMA gradually improves across the three datasets. This aligns with the expectation that providing more auxiliary information allows the model to learn better feature representations and thus enhance its performance on the datasets.
>
> **Q2:** We thank the reviewer for this great question. Our EGMA model is designed based on the CLIP architecture and supports the co-training of both eye-gaze and non-eye-gaze data, which provides it with significant flexibility. Recently, a study has seamlessly integrated eye-tracking technology into ultrasound imaging [8], where ultrasound images, eye-gaze data, and diagnostic texts were collected. EGMA can be easily adapted to this dataset with minimal modifications, primarily involving the replacement of the encoder with one pre-trained on ultrasound images.
>
> References:
>
> [1] Kröger, J.L., et al. (2020). What does your gaze reveal about you? On the privacy implications of eye tracking.
>
> [2] Katsini, C., et al. (2020). The role of eye gaze in security and privacy applications: Survey and future HCI research directions.
>
> [3] Steil, J., et al. (2019). Privacy-aware eye tracking using differential privacy.
>
> [4] Liu, A., et al. (2019). Differential privacy for eye-tracking data.
>
> [5] Stember, J.N., et al. (2020). Integrating eye tracking and speech recognition accurately annotates MR brain images for deep learning: proof of principle.
>
> [6] Khosravan, N., et al. (2019). A collaborative computer aided diagnosis (C-CAD) system with eye-tracking, sparse attentional model, and deep learning.
>
> [7] Ma, C., et al. (2023). Eye-gaze-guided vision transformer for rectifying shortcut learning.
>
> [8] Men, Q., et al. (2023). Gaze-probe joint guidance with multi-task learning in obstetric ultrasound scanning.

---

### Official Review · Reviewer_YZxW · 2024-07-13

**Soundness:** 4
**Presentation:** 4
**Contribution:** 4
**Rating:** 8
**Confidence:** 4

**Summary:**

This paper proposes a cross-modal alignment method that can optionally learn from eye tracking data that is collected together with the speech of radiologists. The proposed Eye-gaze Guided Multi-modal Alignment (EGMA) system consists of losses that optionally incorporate alignment objectives between sentences and gaze regions (expressed as heatmaps) which allows the contrastive learning scheme to out-perform existing vision-language models for radiology, such as MedCLIP. This is in terms of linear evaluation (supervised classification), zero-shot classification, image-to-text retrieval and text-to-image retrieval. The EGMA method also shows highly interpretable cross-modal attention maps and well clustered embeddings (with respect to known ground-truth classes).

**Strengths:**

A key challenge with papers that propose to incorporate eye gaze data into vision-language model training, is the lack of paired training samples. This paper proposes a very clever solution that is clever in two ways: (a) they propose fine-grained alignment between sentence tokens (originating from speech) and scanpath heatmaps corresponding to the sentences, and (b) design their contrastive objectives to allow for gaze supervision to be optional. This results in a rare demonstration of the benefit of gaze data for improving not only performance (as measured by classification or retrieval) but also interpretability.

The paper is also written very well, with sufficient figures and descriptive text - which is a plus.

The presented quantitative results are comprehensive, on several datasets, and consistently out-performing multiple existing state-of-the-arts. The ablation study, in particular, with regards to varying the amount of available gaze data is very interesting. It shows a gradual and (mostly) consistent increase in zero-shot classification performance with an increasing number of gaze-labeled data. The value of EGMA is further supported by the fact fine-tuning SotA models using EGMA typically results in performance improvements (Appendix F).

**Weaknesses:**

Despite the authors’ best efforts, Fig. 2C and Fig. 2D (and the corresponding Sec. 3.2 and 3.3) are hard to understand. In particular, it is unclear to me why the Mean(Max()) reduces features are considered to be “fine-grained” (it rather seems to be an instance-level feature that allows for gaze-free supervision), and why the cross-modality mapping is necessary.

**Questions:**

Please clarify how the Mean(Max()) operation described in Sec. 3.2 enables fine-grained alignment.

I would also like to understand the motivation for the EGM module, in relation to existing prior art.

Thank you.

**Limitations:**

The authors are very forward and clear about the limitation of using or collecting data where gaze and speech are aligned. However, they also mention that medical practises are evolving to allow for this better. In this reviewer’s opinion, the framework proposed in this paper could be easily applied to non-medical areas as well as its design is domain-agnostic.

Privacy or societal concerns are also adequately mentioned.

---

> ### Author Rebuttal · Authors · 2024-08-07
>
> **Q1\&W1: "Please clarify how the Mean(Max())..." "Despite the authors’ best efforts, Fig. 2C and Fig. 2D...allows for gaze-free supervision)"**
>
> We thank the reviewer for this great question and apologize for any confusion caused by casual description in our paper. First, to clarify the role of fine-grained alignment, we have rewritten the Mean(Max()) operation in Sec. 3.2 of main paper as follows:
>
> $$
> \hat{z}^I_k= \frac{1}{m} \sum^{m}_{j=1}[\underset{j}{max}  {(x^{P2S}_k)} ], ~~ 1 \leqslant j \leqslant m   ~~  (1)
> $$
>
> $$
> \hat{z}^T_k= \frac{1}{n} \sum^{n}_{i=1}[\underset{i}{max} {(x^{S2P}_k)} ], ~~ 1 \leqslant i \leqslant n  ~~  (2)
> $$
>
> In the above equations, $x_k^{P2S} \in \mathbb{R}^{n \times m}$ and $x_k^{S2P} \in \mathbb{R}^{m \times n}$ are similarity matrices between local features within a single instance, and these matrices have been optimized with eye-tracking data as shown in Eq. 4 in main paper. We know that CLIP's contrastive loss can align features well within a batch by computing the similarity between global features of the samples. To continue using this contrastive loss on top of instance-level local features, we need the global similarity between the corresponding text and image for each instance. In this work, instead of directly using the global features output by the encoder as CLIP does, we first calculate local feature similarities and then derive global similarity from the local similarity matrix. Specifically, to compute the global similarity between an image and a diagnostic text within a batch, we first calculate the similarity matrix $x_k^{P2S}$ between the local features of image patches and sentence tokens using Eq. 3 in main paper. Then, we take the maximum value of each column of this matrix, meaning each sentence token gets a similarity score with its most related image patch. Finally, we average these values to obtain the global similarity refined by local similarity. The calculation of the fine-grained global similarity from diagnostic text to image within a batch is similar. This discussion and results will be incorporated into the revised version of the paper.
>
>
> **Q2\&W2: "I would also like to understand the motivation for the EGM module..." "why the cross-modality mapping is necessary."**
>
> We thank the reviewer for this great question. In fact, the EGF and EGM modules proposed in our work are similar to the diagnostic process of radiologists. Many studies [1-5] have pointed out that radiologists perform a global search first and then conduct a detailed examination of the local areas when suspicious lesions are found. Firstly, the EGF module of EGMA corresponds to the global search by radiologists, aligning the local image patch features with the individual sentence features. Secondly, EGM uses the key values from the local similarity matrix computed in the first step as weights, focusing the alignment on certain important image patches and texts, akin to the detailed observation stage after identifying suspicious lesions. Thus, by mimicking the real diagnostic behavior of radiologists, our EGMA further enhances the model's multimodal processing capability and diagnostic accuracy. This discussion will be incorporated into the revised version of the paper.
>
>
> **L: "In this reviewer’s opinion, the framework proposed in this paper could be easily applied to non-medical areas as well as its design is domain-agnostic."**
>
> We greatly appreciate the reviewer's suggestion. In fact, thanks to the inherent flexibility of the EGMA model, it is well-suited for multimodal alignment tasks involving natural images, such as in human-robot interaction and control as well as for education/training purposes. Therefore, we believe this is a promising direction for future expansion. Once again, thank the reviewer for this valuable suggestion. This discussion will be incorporated into the revised version of the paper.
>
> References:
>
> [1] Nodine, C.F., et al. (1987). Using eye movements to study visual search and to improve tumor detection.
>
> [2] Swensson, R.G.. (1980). A two-stage detection model applied to skilled visual search by radiologists.
>
> [3] Kahneman, D.. (2003). A perspective on judgment and choice: Mapping bounded rationality.
>
> [4] Kundel, H.L., et al. (2007). Holistic component of image perception in mammogram interpretation: gaze-tracking study.
>
> [5] Drew, T., et al. (2013). Informatics in radiology: what can you see in a single glance and how might this guide visual search in medical images?

---

> > ### Comment · Reviewer_YZxW · 2024-08-12
> >
> > Thank you very much for the detailed and kind answers.
> >
> > I understand the EGF and EGM losses slightly better after reading your explanation. One follow-up question would be whether the EGF loss, as it expects fine-grained feature consistency (specifically, that the max value corresponds to the correct location of interest), would require that the image encoder is pre-trained prior to pre-training. I see in Sec. B.1 that you mention the use of SwinTransformer and BioClinicalBERT as the image and text encoders, respectively, but it is not clear to me whether you begin from the publicly available pre-trained weights.
> >
> > I would think that the EGF loss may not work very well when training from scratch (as taking the max can lead to improper features being selected).
> >
> > In any case, I will retain my rating as I believe the authors have addressed both mine and most other reviewers' concerns. The authors need not respond to me.

---

### Official Review · Reviewer_hPhG · 2024-07-14

**Soundness:** 2
**Presentation:** 2
**Contribution:** 2
**Rating:** 5
**Confidence:** 4

**Summary:**

This paper proposes utilizing eye-gaze information to aid in learning representations from paired images and texts.  The method was validated on four medical datasets, demonstrating the feasibility of integrating gaze information for multi-modal alignment.

**Strengths:**

The idea of using additional information beyond the image-text pairs is interesting.

**Weaknesses:**

First, if eye-gaze coordinates are noisy and contain errors, introducing eye-gazing into the learning process could negatively impact representation learning with image-text pairs. The image-text pairs are well-defined and reliable data resources. In contrast, the quality of eye-gaze information can be highly dependent on the operators, working conditions, and devices. Introducing potentially noisy and/or unreliable information into the learning process is risky and may lead to degraded performance.This paper does not mention and address this issue and the proposed method lacks for dealing with potentially noisy gaze information (could happen.)

Eye-gaze locations may fall on the cardiac areas in both normal and abnormal cases, but the critical consideration in learning is to develop a representation that can distinguish between these situations. Eye-gazes might provide little to no useful information for this learning process, as there is can be no significantly clear distinction between the eye-gaze coordinates in abnormal versus normal cases. Introducing eye-gaze data is not useful for learning to distinguish between these conditions. The only benefit might be identifying the cardiac area, which can be easily captured by conventional contrastive learning without the need for explicitly provided location information.

The writing is poor. The notations and formulas used in the method section are unprofessional. Notations such as COS, Mean, and Max are too casual and should be changed to be more rigorous and formal.

**Questions:**

If the gaze information is not very accurate, for example, if the gazes occasionally shift to locations not exactly on the objects (e.g., cardiac), would using the gaze information harm the learning performance? If not, why, and if yes, how would you handle such an issue?

**Limitations:**

Please see weakness.

---

> ### Author Rebuttal · Authors · 2024-08-07
>
> **W1\&Q1:** We thank the reviewer for these great questions and apologize for any confusion caused by the lack of discussion in our paper. Indeed, noise and errors in eye-gaze data are as common as noise in images, and these can all affect the model's final performance. In this work, the errors in eye-gaze data primarily stem from two factors: involuntary saccades and microsaccades of the radiologists' eyes [1], and subjective fixation errors [2].
>
> Human eye movements can be categorized into two main types: saccades and fixations [1]. Saccades are the rapid movements between different areas of the visual field and are generally considered noise data that do not involve cognitive processes. Fixations, on the other hand, are brief pauses of the eyes on a small area and are regarded as the primary cognitive behavior. Additionally, due to the structure of the eye, fixations are accompanied by microsaccades, causing the fixation point to drift within the fixation area. Thus, microsaccades are also a major source of noise in eye-gaze data. Fortunately, eye-tracking technology has evolved significantly over the past century, and noise reduction techniques for eye-gaze data have become highly advanced. Currently, all commercial eye-tracking devices come with preprocessing software that uses adaptive methods (e.g., [3]) to filter out noise, greatly facilitating the use of eye-tracking technology. In the attached PDF, Figure R1 shows an example of denoising process in MIMIC-EYE dataset.
>
> Additionally, due to variations in radiologists' expertise and cognitive levels, fixation data may contain errors, such as the reviewer mentioned, "occasionally shift to locations not exactly on the objects."
>
> For the minor shift error of fixations, where the duration of incorrect fixations is small compared to the overall duration, their values in the attention heatmap are low, thus having minimal impact on the mlce loss utilized in our work (Eq. 4 in main paper). In our EGMA, EGF and EGM modules (Sec. 3.2 and Sec. 3.3 in main paper) can also filter out this minor errors in the heatmap. For the significant fixation errors, MIMIC-EYE data has been cleaned to address this issue at the first time. Specifically, these data were first reviewed by professionals to exclude inconsistent diagnoses and fixation errors. Moreover, the MIMIC-EYE dataset includes eye-gaze data from six radiologists. Even if one or two radiologists' fixations are incorrect, the correct fixations from the other radiologists under the same semantic context can mitigate the adverse effects. In the attached PDF, Figure R2 shows an example of this compensation.
>
> As the reviewer pointed out, effectively identifying and mitigating errors in radiologists' diagnostic behavior during model training is crucial, and this will be a focus of our future work. This discussion and results will be incorporated into the revised version of the paper.
>
> **W2:** We thank the reviewer for this insightful question and apologize for any confusion caused by the lack of relevant discussion in our paper. In fact, numerous studies have investigated and proven the close relationship between radiologists' gaze behavior, image content, and diagnostic results. For example, studies [4-9] have found that radiologists fixate more on diseased areas than on healthy ones. Moreover, they discovered that novice radiologists, due to their lack of experience, repeatedly look at diseased areas, resulting in even more fixations in these areas than those of expert radiologists. Figure R3 in the attached PDF shows some comparison cases.
>
> Additionally, the abnormal regions with more fixations have higher weights in the attention heatmap. Our EGMA model can leverage these weights, along with image content and text, to learn better disease diagnosis capabilities and feature representations. This discussion and results will be incorporated into the revised version of the paper.
>
> **W3:** We are grateful for the reviewer’s attentive analysis and helpful feedback. We have revised the formulas as follows: (1) Change "COS" (Eq. 1 and Eq. 3 in main paper) to lowercase, as shown in Equations 1 and 2 below:
> $$
>     s_{k,l}^{I2T} = cos(z_k^I, z_l^T), \ s_{k,l}^{T2I} = cos(z_k^T, z_l^I) \quad  1 \leqslant l \leqslant b  ~~ (1) \\
> $$
>
> $$
>     x_k^{S2P} = cos(S_k^j, P_k^i), \ x_k^{P2S} = cos(P_k^i, S_k^j)   ~~ (2)
> $$
> (2) Change "Mean/Max" (Sec. 3.2 in main paper) to the following Equations 3 and 4:
> $$
> \hat{z}^I_k= \frac{1}{m} \sum^{m}_{j=1}[\underset{j}{max}  {(x^{P2S}_k)} ], ~~ 1 \leqslant j \leqslant m  ~~ (3)
> $$
>
> $$
> \hat{z}^T_k= \frac{1}{n} \sum^{n}_{i=1}[\underset{i}{max} {(x^{S2P}_k)} ], ~~ 1 \leqslant i \leqslant n  ~~ (4)
> $$
> These revised formulas will be incorporated into the revised version of the paper.
>
> References:
>
> [1] Chamberlain, L.. (2007). Eye tracking methodology: theory and practice.
>
> [2] Brady, A.P.. (2017). Error and discrepancy in radiology: inevitable or avoidable?
>
> [3] Nyström, M. and Holmqvist, K.. (2010). An adaptive algorithm for fixation, saccade, and glissade detection in eyetracking data.
>
> [4] Nodine, C.F., et al. (1996). Nature of expertise in searching mammograms for breast masses.
>
> [5] Mallett, S., et al. (2010). Tracking eye gaze during interpretation of endoluminal three-dimensional CT colonography: visual perception of experienced and inexperienced readers.
>
> [6] Voisin, S., et al. (2013). Investigating the association of eye gaze pattern and diagnostic error in mammography.
>
> [7] Giovinco, N. A., et al. (2015). A passing glance? Differences in eye tracking and gaze patterns between trainees and experts reading plain film bunion radiographs.
>
> [8] Van der Gijp, A., et al. (2017). How visual search relates to visual diagnostic performance: a narrative systematic review of eye-tracking research in radiology.
>
> [9] Tourassi, G., et al. (2013). Investigating the link between radiologists' gaze, diagnostic decision, and image content.

---

> > ### Comment · Reviewer_hPhG · 2024-08-11
> >
> > I have read the authors' rebuttal which has addressed my concerns to some degree. I have raised my rating accordingly. Thanks.

---

> > > ### Author Response · Authors · 2024-08-12
> > >
> > > We appreciate the timely response. We are pleased to know that we have successfully addressed your concerns.
> > >
> > > Thanks for engaging with our work and adjusting your evaluation.

---

### Author Rebuttal · Authors · 2024-08-07

We would like to thank all reviewers for their careful reading, valuable comments, and recognition of the contributions of our work. We have provided itemized responses to the questions and suggestions from the reviewers. We are also pleased to receive the positive feedbacks from reviewers, particularly:

1. Novel and interesting method [Reviewers hPhG, YZxW, V2ie, and n21y].

2. Thorough evaluations and ablation studies [Reviewers YZxW, V2ie, and n21y].

3. Potential for broader application [Reviewers V2ie and n21y].

4. Excellent and clear written [Reviewer YZxW].

We found that reviewers are particularly concerned about the general availability and quality of the eye-gaze data. Here are brief responses to these two concerns:

**Data availability issue [Reviewers V2ie and n21y].**

Multiple studies [1-5] have investigated the cognitive behavior in radiologists' eye-gaze data, demonstrating its close relationship with images and diagnostic results. This highlights the availability and value of eye-gaze data in the medical field. In addition, with the growing research and solution development in the practical application of eye-tracking systems in the medical field [6-13], we envision that both the quality and the availability of eye-gaze data will be greatly improved. The recent success of eye-tracking technology in commercial applications (e.g., Apple Vision Pro, eye-tracking in iOS 18) further highlights its core role as a next-generation human-computer interaction technology. In our work, the model's zero-shot capability on other datasets significantly improved after eye-gaze guidance. This demonstrates that the model has learned more generalizable feature representation through the use of eye-gaze data. Furthermoer, we conducted ablation experiments (Tab. 4 in the main paper) to verify that our model can benefit from very limited eye-gaze data. Therefore, the ability to efficiently utilize eye-gaze data can provide valuable assistance to the model both now and future. Detailed explanations are provided in response to the respective questions from the reviewers (**Reviewer V2ie**:W4, **Reviewer n21y**:W).

**Data quality issue [Reviewers hPhG and n21y].**

Eye-tracking technology has evolved significantly over the past century, and noise reduction techniques for eye-gaze data have become highly advanced [14-16]. We also included examples in the attached PDF (Figure R1 and Figure R2) to illustrate how our model can mitigate the impact of noise and errors in the eye-gaze data. Detailed explanations are provided in response to the respective questions from the reviewers (**Reviewer hPhG**:W1\&Q1, **Reviewer n21y**:Q1).


References:

[1] Nodine, C.F., et al. (1987). Using eye movements to study visual search and to improve tumor detection.

[2] Swensson, R.G.. (1980). A two-stage detection model applied to skilled visual search by radiologists.

[3] Kahneman, D.. (2003). A perspective on judgment and choice: Mapping bounded rationality.

[4] Kundel, H.L., et al. (2007). Holistic component of image perception in mammogram interpretation: gaze-tracking study.

[5] Drew, T., et al. (2013). Informatics in radiology: what can you see in a single glance and how might this guide visual search in medical images?

[6] Khosravan, N., et al. (2019). A collaborative computer aided diagnosis (C-CAD) system with eye-tracking, sparse attentional model, and deep learning.

[7] Stember, J.N., et al. (2020). Integrating eye tracking and speech recognition accurately annotates MR brain images for deep learning: proof of principle.

[8] Wang, S., et al. (2022). Follow my eye: Using gaze to supervise computer-aided diagnosis.

[9] Ma, C., et al. (2023). Eye-gaze-guided vision transformer for rectifying shortcut learning.

[10] Men, Q., et al. (2023). Gaze-probe joint guidance with multi-task learning in obstetric ultrasound scanning.

[11] Ma, C., et al. (2023). Rectify vit shortcut learning by visual saliency.

[12] Ji, C., et al. (2023). Mammo-net: Integrating gaze supervision and interactive information in multi-view mammogram classification.

[13] Zhao, Z., et al. (2024). Mining gaze for contrastive learning toward computer-assisted diagnosis.

[14] Salvucci, D.D. and Goldberg, J.H.. (2000). Identifying fixations and saccades in eye-tracking protocols.

[15] Smeets, J.B. and Hooge, I.T.. (2003). Nature of variability in saccades.

[16] Nyström, M. and Holmqvist, K.. (2010). An adaptive algorithm for fixation, saccade, and glissade detection in eyetracking data.

---

### Decision · Program_Chairs · 2024-09-25

**Decision:**

Accept (poster)

**Comment:**

This paper introduces an eye gaze guided vision language learning to improve visual and text representation learning for medical applications .  It receives one strong accept , one accept, and two borderline accepts . The reviewers in general like this work, finding its ideas using gaze to guide learning novel, papers well written, with extensive experiments that demonstrate the superior performance of their method.  They have some concerns about data quality, data accessibility, and data privacy.  The authors rebuttal is extensive and well addresses the reviewers concerns .  I hence recommend its acceptance.